# Improved Particle Swarm Optimization Based on Entropy and Its Application in Implicit Generalized Predictive Control

**DOI:** 10.3390/e24010048

**Published:** 2021-12-27

**Authors:** Jinfang Zhang, Yuzhuo Zhai, Zhongya Han, Jiahui Lu

**Affiliations:** School of Control and Computer Engineering, North China Electric Power University, Beijing 102206, China; zyz@ncepu.edu.cn (Y.Z.); hzy@ncepu.edu.cn (Z.H.); 120202227029@ncepu.edu.cn (J.L.)

**Keywords:** implicit generalized predictive control, system entropy, particle swarm optimization, reverse optimization strategy, inertia weight

## Abstract

Setting sights on the problem of input-output constraints in most industrial systems, an implicit generalized predictive control algorithm based on an improved particle swarm optimization algorithm (PSO) is presented in this paper. PSO has the advantages of high precision and fast convergence speed in solving constraint problems. In order to effectively avoid the problems of premature and slow operation in the later stage, combined with the idea of the entropy of system (SR), a new weight attenuation strategy and local jump out optimization strategy are introduced into PSO. The velocity update mechanism is cancelled, and the algorithm is adjusted respectively in the iterative process and after falling into local optimization. The improved PSO is used to optimize the performance index in predictive control. The combination of PSO and gradient optimization for rolling-horizon improves the optimization effect of the algorithm. The simulation results show that the system overshoot is reduced by about 7.5% and the settling time is reduced by about 6% compared with the implicit generalized predictive control algorithm based on particle swarm optimization algorithm (PSO-IGPC).

## 1. Introduction

In the development of the industrial field in recent years, predictive control has been widely used in IT, electric power, aerospace, automobile, and other fields because of its wide application range of predictive model, rolling-horizon, and good control effect [1,2,3,4]. In 1987, Clarke and Mohtadi propose the generalized predictive control algorithm (GPC) [5]. GPC retains the essential characteristics of model algorithm control (MAC) and dynamic matrix control (DMC). Its model adopts the controlled auto regressive integrated moving average (CARIMA), and combines adaptive control with predictive control to eliminate the output error of the predictive model caused by slow time-varying process parameters. Although adaptive control technology is widely valued in the industry, it requires high model accuracy. GPC not only combines the characteristics of online identification of adaptive control and is suitable for stochastic systems, it also retains the rolling-horizon part of predictive control and widely applicable models [6]. Based on GPC, implicit generalized predictive control (IGPC) eliminates the solution of the Diophantine equation and avoids the problems of large amounts of calculation and cumbersome processes [7]. This makes IGPC more widely used in the production process.

In terms of rapidity, robustness, and stability, the control result of GPC is more notable than that of traditional PID control [8,9]. Nowadays, most scholars improve GPC in two aspects: The establishment of the prediction model and the improvement of the optimization method. Cheng [10] enhances the control effect by combining fuzzy control modeling and predictive control, but the problems pointed out in this paper still cannot be solved. R.C and T.E [11] obtain the controlled object model through the slope algorithm of second-order difference, but it is only suitable for the SISO system. Chen [12] presents a modeling method through the frame-cum-rate adjustment (FRA) algorithm and bat algorithm (Ba) to replace the traditional least square method. Zheng [13] combines the ant colony optimization algorithm with predictive control and compares it with PID control. Wu [14] combines ADRC with GPC to form a new controller and simplifies the original system to a cascade-integrator. Jiale [15] combines PD and IGPC to form a new algorithm, which selects the control variables through the output of the system, rather than a variable calculated from the predicted value of the delay. In reference [16], PSO is added to IGPC to form a new algorithm to solve the control problem with constraints in industrial systems. The author calls this algorithm AMPSO-IGPC.

Nowadays, most scholars combine intelligent algorithms with GPC to improve it. Among many swarm intelligence algorithms, the PSO has been widely used because of its simple structure and the optimization effect is good under constraints, including power, AI intelligence, biology, Internet of Things, aerospace, and other fields [17,18,19,20,21]. In 1995, Eberhart and Kennedy [22] proposed PSO by simulating the process of social animals sharing location information to achieve the goal. Shi [23] analyzes the influence of weight change, reduces the weight in equal proportion, and improves the convergence of the algorithm. Guo [24] introduces the nonlinear decreasing weight strategy, which can enter the local search faster by increasing the decrease rate of weight at an early stage of the algorithm. However, the author found that the improvement of weight is limited to the improvement of control effect. Cai [25] adds the subgroup strategy to remove the disadvantage of long adjustment time of PSO and cites the restart operator mechanism to improve the algorithm. Hayashida adds the subgroup into PSO [26]. The simulation results exhibit that it has high performance in finding the optimal solution to multi-dimensional and nonlinear problems. However, the optimization of high-order functions is still not accurate enough.

At present, different methods have improved PSO convergence in varying degrees, but there are still some problems, such as insufficient accuracy and poor effect, and most of them still retain the update of speed term. Regarding the above problems, this paper combines the idea of the entropy of the system (SR) with PSO, presenting a new weight strategy and threshold judgment mechanism to improve PSO on the basis of removing the influence of speed term. For the sake of brevity, we call the entropy of the system SR. We can regard the particle and search space of PSO as a system and analyze PSO from the change of entropy in the system. In the process of PSO optimization, the SR changes from time to time. By disturbing the system of PSO, the entropy of the algorithm increases in the optimization process to avoid and get rid of local optimization, so as to improve the effect.

Aiming at the problems of premature and low optimization accuracy of high-dimensional function in the above PSO, it is improved in the second chapter. Considering that there are many constraints in the actual industrial process, and the traditional GPC usually finds the local optimal solution when solving such problems [27,28], to improve the control effect, the author combines the PSO with the implicit generalized predictive control. It improves the GPC control effect by enhancing PSO optimization near the constraint limit to form a new control algorithm, which is called SPPSO-IGPC. The specific steps of the algorithm are described in Section 3. The control effect is verified by simulation in Section 4.

## 2. Improved Particle Swarm Optimization Algorithm

The principle of PSO is introduced in Section 2.1. Due to the problems of premature and low optimization accuracy of a high-dimensional function, the effect of combining PSO with GPC is not good. In the process of optimization, PSO is improved by the methods shown in Section 2.2 and Section 2.3 to reduce the occurrence of premature. After precocity, the threshold judgment strategy shown in Section 2.4 is added to adjust it. The simulation results are shown in Section 2.5.

### 2.1. Particle Swarm Optimization Algorithm

There are a limited number of M particles in the population. Each of them has two attributes: Position, P and velocity, V. In each iteration, the particle will move towards the optimal position of its nearby area, and the velocity and position will be updated. The particle search space is D-dimensional. The position information of particle “*i*” is expressed as $k_{id}$ and the velocity information is $v_{id}$. The individual extreme value of each particle is called $g_{id}$, and the extreme value of the entire population is called $g_{gd}$. In each iteration, the current fitness of a particle will be compared with its individual optimal fitness, the better one will be selected for updating, and then the optimal individual extreme value in the population will be selected to compare and update the global optimal. The formula of PSO [29] first proposed by Kennedy and Clerc is as follows:(1)$$v_{id}(t+1)=\omega v_{id}(t)+c_{1}rand(g_{id}(t)-k_{id}(t))+c_{2}rand(g_{gd}(t)-k_{id}(t))$$
(2)$$k_{id}(t+1)=k_{id}(t)+v_{id}(t+1).$$

In Formulas (1) and (2), ω is the inertia weight; i=1,2,……,m; rand is a natural number that varies randomly between 0 and 1; $c_{1}$ and $c_{2}$ are learning factors, and $c_{1}=c_{2}=2$ is generally taken according to experience; velocity update is limited to $v_{id}\in [v_{id}\min,v_{id}\max]$; and the location update is limited to $k_{id}\in [k_{id}\min,k_{id}\max]$.

### 2.2. Remove the Influence of Velocity Term

PSO imitates the foraging activities of birds. Each “bird” constantly updates the velocity and position, as shown in Equations (1) and (2), but the position update is only superimposed with the velocity term, which cannot well reflect the actual process of “foraging”. After analysis, it is found that the velocity term is unnecessary. The process of each particle optimization is to make the position term approach the optimal value infinitely, and the velocity represents the “moving direction”, which also brings the possibility of being far away from the destination, that is, divergence. The velocity term is removed and the position is updated directly, which improves the convergence velocity and accuracy to a certain extent. Firstly, preprocess Equations (1) and (2), make $a=c_{1}r_{1}$, $b=c_{2}r_{2}$ and $d=\frac{ag_{id}+bg_{gd}}{a+b}$, and write Equations (1) and (2) as follows:(3)$$v_{id}(t+1)=\omega v_{id}(t)+(a+b)(d-k_{id}(t))$$
(4)$$v_{id}(t+1)=k_{id}(t+1)-k_{id}(t).$$

From Equations (3) and (4):(5)$$k_{id}(t+2)=(1-a-b)k_{id}(t+1)+\omega v_{id}(t+1)+d(a+b).$$

After sorting:(6)$$k_{id}(t+2)+(a+b-\omega -1)k_{id}(t+1)+\omega k_{id}(t)-d(a+b)=0.$$

It can be seen from Equation (6) that the velocity term has no influence on the algorithm, and the influence of the velocity term constraint on the optimization of the algorithm is eliminated after being removed.

### 2.3. Weight Attenuation Strategy Combined with SR

Entropy represents the degree of chaos of the system, that is, the greater the degree of chaos of the system, the greater the entropy. We can regard the particle and search space of PSO as a system and analyze PSO from the change of entropy in the system. In the process of PSO optimization, the SR changes from time to time. When the particle falls into the local optimum, the SR reaches a very small value. According to the principle of minimum entropy, if the population has no strong external interference, the total entropy of the system is always decreasing and the order is increasing. From this point of view, if the particle wants to jump out of local optimization, we can impose an interference on the system to increase its total entropy and enhance the degree of chaos.

In the process of population particle optimization, the inertia weight coefficient ω affects the particle search effect and optimization accuracy. When ω is large, the corresponding global search ability is strong, and when ω is small, the corresponding local search is more accurate. Most scholars’ improvement on weight is to make it decrease in medium proportion in the iterative process, as shown in Figure 1 (w1). In Figure 1, the horizontal axis is time and the vertical axis is the value of weight. First, conduct a global and large-scale search, and then reduce the accuracy in the later stage of the iteration. Some adopt the weight decreasing strategy of concave function or convex function. When the number of iterations increases, the weight reduction speed slows down, as shown in Figure 1 (w2 and w3).

However, it does not bring much improvement to the problem of PSO falling into local optimization. Aimed towards the problem, combined with the concept of SR, this paper adds the idea of intermediate enhancement in the process of weight attenuation, that is, the strategy of rising again in the middle of the iteration, showing the trend of combining concave function and convex function, as shown in Figure 1 (w4). A disturbance is added in the middle of the iteration to increase the entropy and chaos of the system. Its function is to carry out a global search at the maximum speed in the initial stage, and give full play to the early advantages of PSO as much as possible. In the middle of the iteration, some particles will enter the local optimum, and the change of position item is small. At this time, increase the proportion of global search to make the particles jump out of the local wandering state and search for the optimal value again. Finally, the weight is reduced to the minimum and the local search accuracy is improved to the maximum. The relationship between weight and iteration times is as follows:(7)$$\omega =\omega_{\max}+(\frac{1}{1+e^{-\frac{6T+12t}{T}}}-\frac{2t}{T})(\omega_{\max}-\omega_{\min}).$$

In the formula, $\omega \in [\omega_{\min},\omega_{\max}]$ and *t* are the current iteration population times, *T* is the maximum population iteration times, and *e* is the natural constant. According to experience, when ω is 0.8, it can reach the global optimal position faster. In this paper, $\omega_{\min}=0$ and $\omega_{\max}=0.8$ are taken.

### 2.4. Local Optimal Judgment Threshold

Usually, PSO will fall into local optimization in the process of optimization, which makes the control effect worse or unable to meet the requirements. At this time, the SR becomes smaller and the system is more orderly. In order to enhance the global search ability, that is, increase the SR, Section 2.2 and Section 2.3 are measures to reduce this local phenomenon in the iterative process, and this section introduces the judgment threshold. In the population iteration process, when the number of times that the individual optimal value of a particle stops updating exceeds the threshold, the active position update mechanism is implemented to make the particle move to the historical position vector and the opposite direction for re-optimization and to increase the degree of confusion and the SR, as shown in Figure 2. This method aims to determine the particle moving state and jump out of the local optimum, so as to enhance the performance of the algorithm.

Assuming that ① to ③ in the figure is the trajectory of a particle in turn, the individual optimal value $g_{id}$ of the particle after each position update is compared and updated. Due to the uncontrollability of the nonlinear system, the scope of this mechanism is set in the whole iterative process so as to maintain the global effect in the early stage and the local accuracy in the later stage. If $g_{id}$ for four consecutive times is the same as the previous time, it means that the particle falls into the local optimal state. At this time, the particle position is updated again. As shown in the figure, the current motion position forcibly jumps out of the local search according to the vector and reverse motion of the historical three motion positions to keep the particle active. The formula of particle local determination and position update mechanism is as follows:(8)$$th_{id}(t)=\left\{ \begin{array}{l} 1{\;g}_{id}(t)=g_{id}(t-1)\\ 0{\;g}_{id}(t)\neq g_{id}(t-1) \end{array}\right..$$

In the formula, $th_{id}(t)$ is the position judgment value of the “*i*” particle at time *t*. When the individual optimal value is the same as the previous time, the judgment value is one; otherwise, it is zero. When the cumulative value of four consecutive judgment values is equal to the threshold, forced position update:(9)$$\sum_{x=t-2}^{x=t}th_{id}(x)=4$$
(10)$$k_{id}(t+1)=k_{id}(t)-\frac{3(k_{id}(t+2)-k_{id}(t))}{4}$$
$$th_{id}(t)=0.$$

After the position is updated, set the judgment value to zero. The location update itself is random, so the random factor is no longer added.

### 2.5. Simulation Analysis

The optimization ability of the improved PSO needs to be tested separately. The test is divided into four different algorithms: PSO, PSO with new weight strategy (wPSO), PSO with velocity term removed and weight strategy added (wdPSO), and through the improved particle swarm optimization (SPPSO) in Section 2. Next, the four algorithms will be compared and tested through six high-dimensional test functions to show their optimization efficiency and accuracy. We have carried out 30-dimensional, 50-dimensional, and 80-dimensional tests respectively, and found that the results can achieve the optimization effect. In order to highlight the effect of the algorithm, this paper shows the test results under 80 dimensional.

Among them, the number of iterations M=1000, population particle N=60, dimension D=80, and unimodal functions are shown in F1, F2, and F3 of Table 1. And F4, F5, and F6 are multimodal functions.

To verify the universality of the simulation results, more than 20 tests have been carried out for each test, and the optimal value corresponding to the fitness function is shown in Table 2. In the table, X represents the number of iterations for the algorithm to find the best value.

We can clearly see that the first two algorithms are not good at optimizing high-dimensional functions, and the changing trend of the global optimal value is shown in Figure 3. For the unimodal function, it can be seen that the convergence speed of PSO and wPSO is relatively slow, but the optimization accuracy of wPSO will be better than that of PSO. By increasing the SR, changing the weight and jumping out of the local optimization, we can achieve a better optimization effect. The results of wdPSO and SPPSO are similar, which shows that the improvement of the algorithm is more obvious by removing the speed term and retaining only the position term.

In the multimodal function, as shown in Figure 4a,c, wdPSO also has the same problem, while SPPSO also maintains good optimization speed and accuracy. It jumps out of the premature situation through the setting of the threshold, and the effect is better than other algorithms.

## 3. Implicit Generalized Predictive Control Algorithm-Based SPPSO

Section 3.1 first introduces the principle of GPC. Due to the poor control effect of GPC under constraints, this paper adds the improved algorithm SPPSO to IGPC for joint optimization. The specific steps and schematic diagram are shown in Section 3.2.

### 3.1. Generalized Predictive Control Algorithm

GPC includes prediction model, rolling-horizon, feedback correction, and other modules.

#### 3.1.1. Prediction Model

The controlled object of GPC is usually described by the CARIMA model:(11)$$A(z^{-1})y(t)=B(z^{-1})u(t-1)+\frac{C(z^{-1})\omega (t)}{\Delta }.$$

In the formula, $A(z^{-1})$, $B(z^{-1})$, and $C(z^{-1})$ are the polynomial of *B*. The difference operator is $\Delta =1-z^{-1}$. For the convenience of calculation, the delay of the system is assumed to be d=1. Of course, just make the coefficient of the first d−1 term of polynomial $B(z^{-1})$ equal to zero if d>1 [30]. ω(t) is a white noise sequence. In order to highlight the advantages of the algorithm, let us make $C(z^{-1})=1$. So as to derive the predicted value after step J, the Diophantine equation is introduced to solve it:(12)$$1=E_{j}(z^{-1})A(z^{-1})\Delta +z^{-j}F_{j}(z^{-1})$$
(13)$$G_{j}(z^{-1})=E_{j}(z^{-1})B(z^{-1})={\bar{G}}_{j}(z^{-1})+z^{-(j-1)}H_{j}(z^{-1}).$$

Multiply $E_{j}(z^{-1})(z^{j}-z^{j-1})$ at both ends of Equation (11) and ignore the future time disturbance. According to Equation (12), the output prediction value at t+j time is expressed as follows:(14)$$\bar{y}(t+j|t)={\bar{G}}_{j}(z^{-1})\Delta u(t+j-1|t)+H_{j}(z^{-1})\Delta u(t)+F_{j}(z^{-1})y(t).$$

#### 3.1.2. Rolling-Horizon

The form of performance index at the specified *t* time is as follows:(15)$$\min J(t)=E\left\{\sum_{j=1}^{n}{\left[y(t+j|t)-\omega (t+j)\right]}^{2}+\sum_{j=1}^{m}\lambda {\left[\Delta u(t+j-1|t)\right]}^{2}\right\}$$
(16)$$\omega (t+j)=\sigma^{j}y_{r}(t+j-1)+(1-\sigma^{j})y_{r}\;(j=1,2,\ldots ,n).$$

Among them, ω(t+j) is the reference track; $y_{r}$ is the set value; E is the mathematical expectation; *n* and *m* are the prediction and control time domains, respectively, that is, the control quantities are equal after step *m*; and λ and σ are the weighting coefficient and softening factor respectively, σ∈[0,1). According to Formula (14), Equation (15) is written as a vector in the form of:(17)$$J(t)={\left(\bar{y}-\omega \right)}^{T}(\bar{y}-\omega )+\Delta u^{T}\lambda \Delta u.$$

In Formula (17), ω=[ω(k+1),……,ω(k+n)]. When $G^{T}G+\lambda I$ is a nonsingular matrix, the optimal solution of Equation (17) is:(18)$$\Delta u={\left(G^{T}G+\lambda I\right)}^{-1}G^{T}(\omega -f)$$
(19)u(t)=u(t−1)+Δu(t)
(20)$$f=H_{n}(z^{-1})\Delta u(t)+F_{n}(z^{-1})y(t).$$

### 3.2. Improved Particle Swarm Optimization-Based IGPC

Because the actual industrial production often has constraint problems, GPC cannot meet the control effect, so the author introduces PSO, and improves the problems of premature, slow optimization, and falling into local optimization when PSO is used to deal with high-dimensional systems. It is found that the optimization effect of GPC at the “boundary” is not ideal when dealing with the problem with constraints. Therefore, the PSO is combined with the gradient optimization of predictive control, and the algorithm is improved in the optimization process and after judgment. Due to the large amount of calculation and the cumbersome process of solving the Diophantine equation, the implicit generalized predictive control algorithm (IGPC) is selected in this paper. When $\frac{1}{2}\Delta u_{\min}<\Delta u<\frac{1}{2}\Delta u_{\max}$, the gradient optimization of GPC is used to solve the optimal control rate. When $\Delta u_{\min}<\Delta u<\frac{1}{2}\Delta u_{\min}$ or $\frac{1}{2}\Delta u_{\max}<\Delta u<\Delta u_{\max}$, the improved PSO is used for optimization. Figure 5 is its schematic diagram. Take the performance index of rolling-horizon as the fitness function of PSO, which is Formula (15):$$\min J(t)=E\left\{\sum_{j=1}^{n}{\left[y(t+j|t)-\omega (t+j)\right]}^{2}+\sum_{j=1}^{m}\lambda {\left[\Delta u(t+j-1|t)\right]}^{2}\right\}.$$

Firstly, the reference trajectory is calculated through the given value, and then the feedback correction is carried out in combination with the difference between the predicted output and the actual output, so as to calculate the control increment at the next time and output it to the controlled object. The steps of improving the algorithm are shown in Table 3.

The system flow chart is shown in Figure 6.

## 4. Simulation Study

Aiming towards the problems with constraints in actual industrial production, this paper combines the improved PSO with IGPC for predictive control. In the control effect test, it is compared with IGPC, PSO-IGPC, and AMPSO-IGPC in the literature [16], respectively. The results are as follows. The vehicle engine system model studied in reference [7]:$$y(k)-1.496585y(k-1)+0.496585y(k-2)=0.5u(k-2)+\frac{\xi (k)}{\Delta }.$$

IGPC parameter settings: $y_{r}$ is a square wave signal; n=6; m=2; λ=0.6; σ=0.35; and $\Delta u_{\min}=-1$, $\Delta u_{\max}=1$.

SPPSO parameter settings: Learning factor $c_{1}=c_{2}=2$; $\omega_{\min}=0$, $\omega_{\max}=0.8$; the population particle number is 40; and the number of iterations is 100.

In Figure 7a,b, the overshoot of the control system is reduced after the combination of PSO and IGPC. Still, the settling time and fluctuation range of the control rate do not change, the overall effect is poor and the improvement is limited. However, in industrial production, the system is accompanied by unknown interference at any time, and the adjustment time needs to be reduced as much as possible. If the control system becomes a high-dimensional system, the control effect of PSO-IGPC will not meet the requirements after adding interference. The simulation is carried out according to the method provided in document [16], and the results are shown in Figure 7c. AMPSO-IGPC can reduce the overshoot and enhance the ability to track the given value. However, the fluctuation of control increment is still obvious. After combining SPPSO with GPC, it can be seen in Figure 7a,d that the overshoot of the control system is reduced, about 7.5% less than that of the traditional IGPC. The settling time is shortened by about 6%, the fluctuation of the control rate is reduced, and the control effect is excellent. This proves that our improvement is effective and can deal with a series of problems about real-time and randomness in the actual production environment.

## 5. Conclusions

So as to deal with the control problem of the generalized predictive control algorithm under constraints, PSO is introduced to improve. In the research process, combined with the idea of SR, the premature problem in a high-dimensional environment is processed and improved accordingly. On the basis of removing the speed term, the weight attenuation strategy combined with SR and the threshold judgment mechanism are added, respectively. The improved algorithm is simulated with MATLAB software and compared with other algorithms. The results show that the optimization accuracy and time of SPPSO are better, and the performance is better in a high-dimensional environment. Then IGPC and SPPSO are combined for predictive control. In the simulation example, it can be seen that the control effect is improved, and is faster and smoother than the traditional PID control or GPC control. In future studies, the algorithm will be applied to more complex working conditions to highlight its advantages.

## Figures and Tables

**Figure 1 entropy-24-00048-f001:**
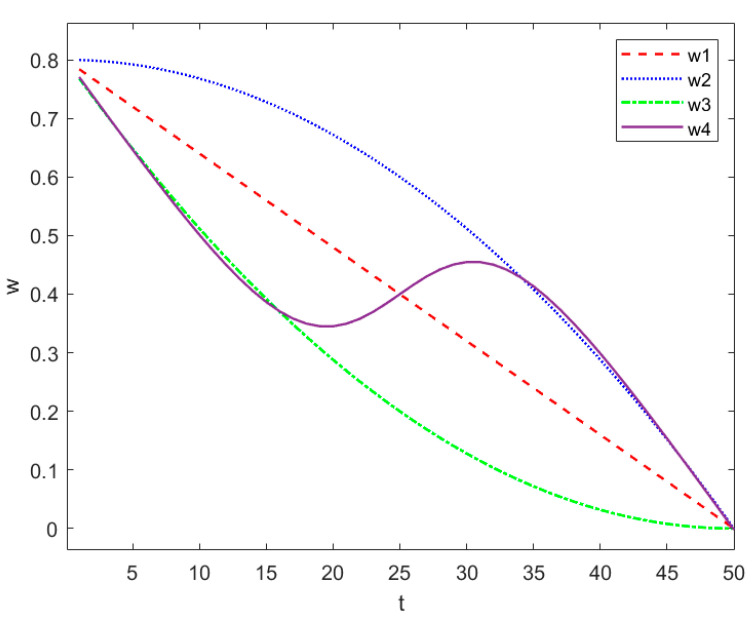
Schematic diagram of weight change.

**Figure 2 entropy-24-00048-f002:**
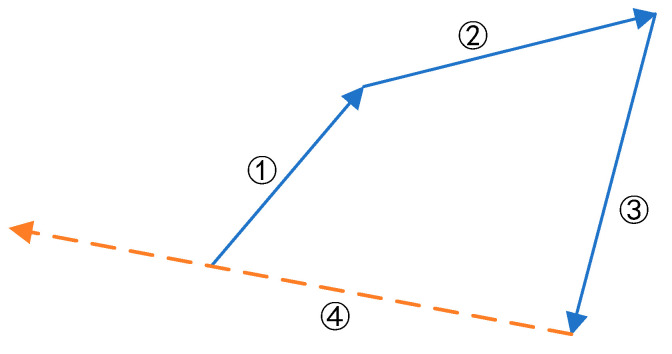
Schematic diagram of particle position update.

**Figure 3 entropy-24-00048-f003:**
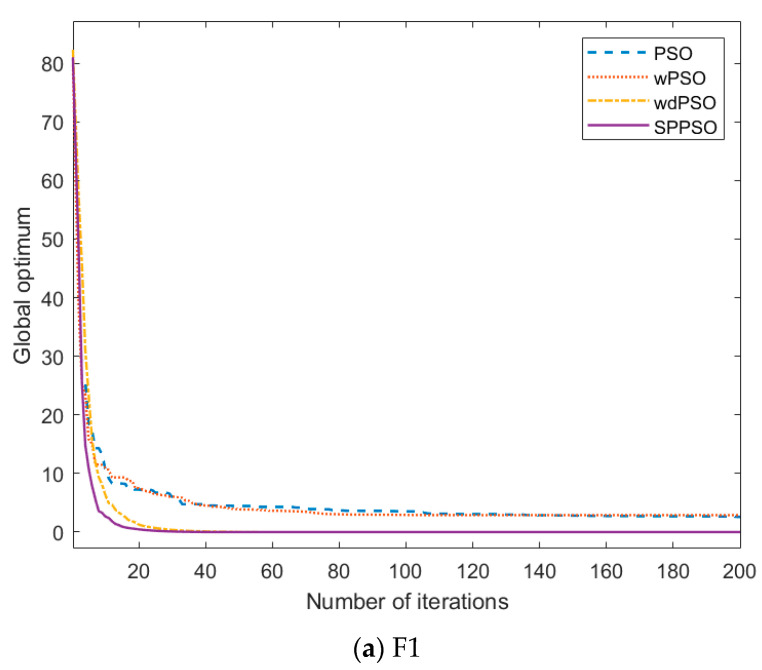
Variation of the global optimal value of the unimodal function.

**Figure 4 entropy-24-00048-f004:**
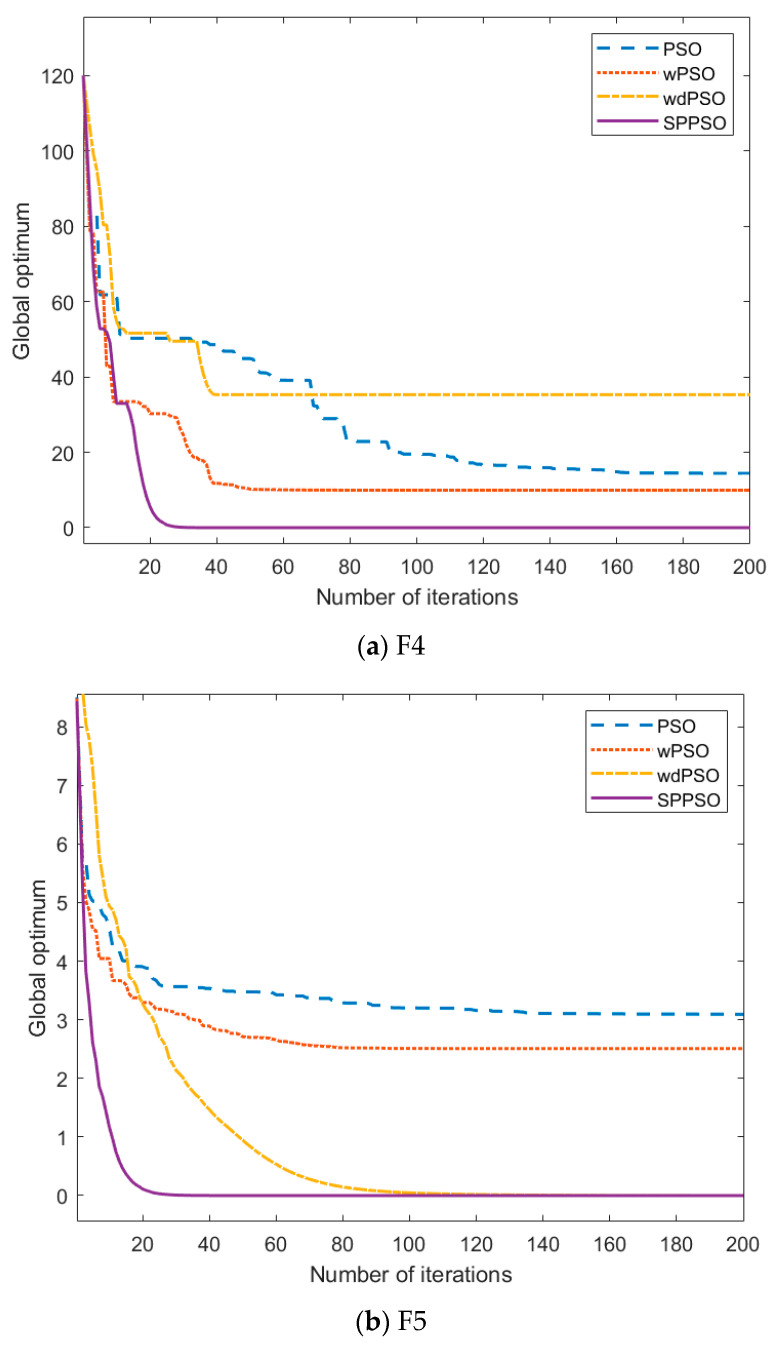
Change of the global optimal value of multimodal function.

**Figure 5 entropy-24-00048-f005:**
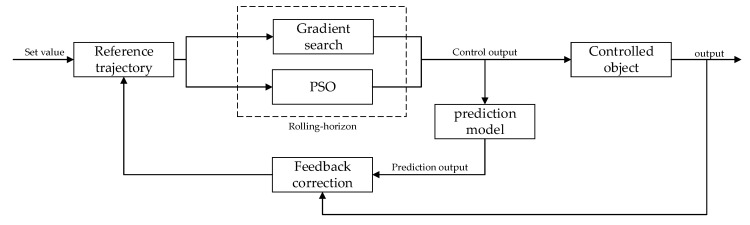
Schematic diagram of GPC.

**Figure 6 entropy-24-00048-f006:**
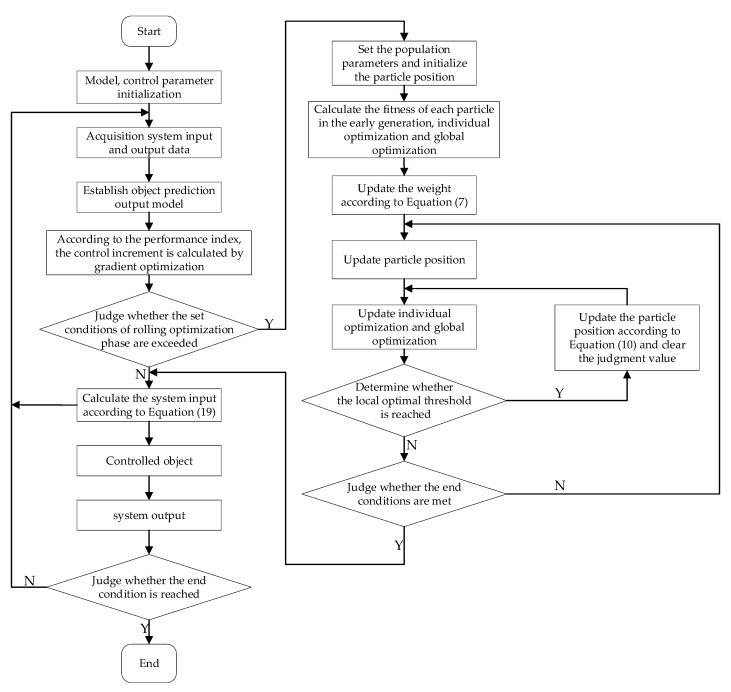
SPPSO-IGPC flow chart.

**Figure 7 entropy-24-00048-f007:**
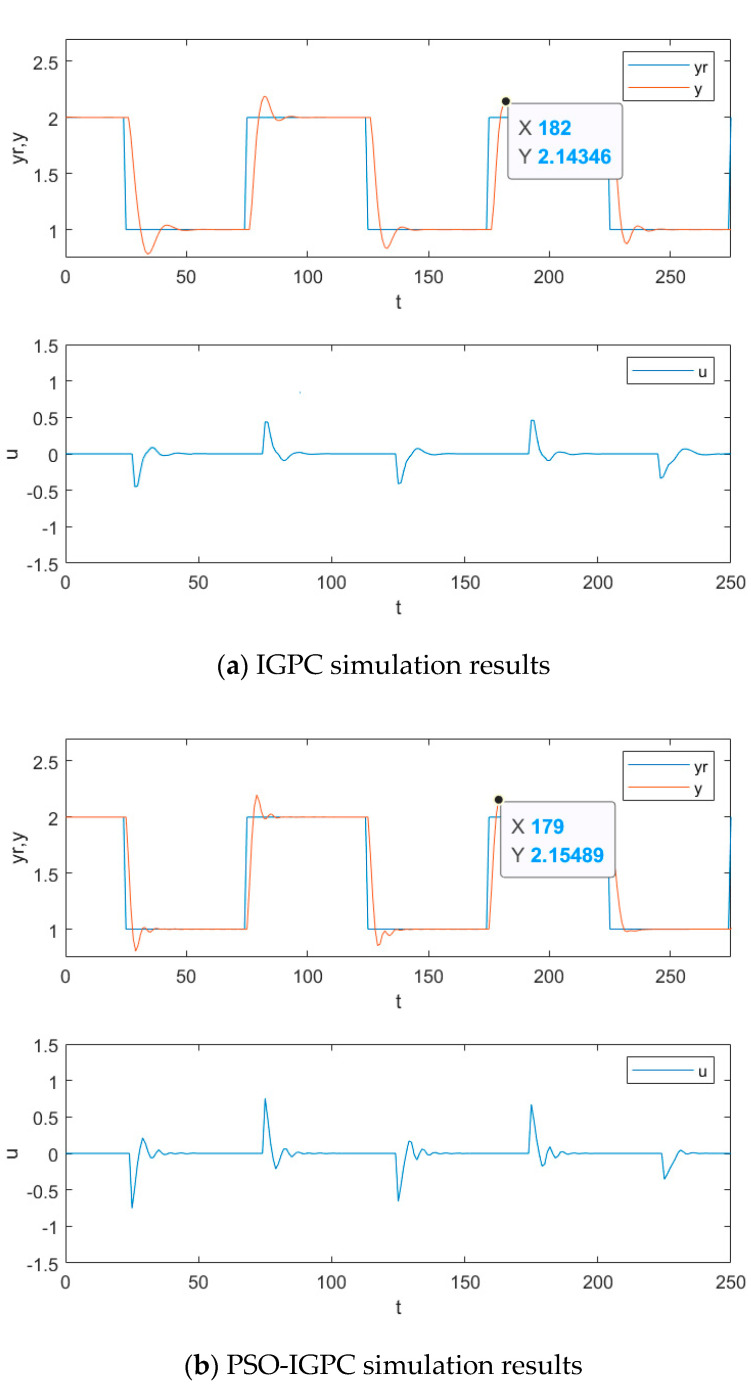
Simulation diagram of the control effect.

**Table 1 entropy-24-00048-t001:** Test function.

Function Name	Function
F1	$F_{1}(x)=\sum_{i=1}^{m}x_{i}^{2}$
F2	$F_{2}(x)=\prod_{i=1}^{m}\left|x_{i}\right|+\sum_{i=1}^{m}\left|x_{i}\right|$
F3	$F_{3}(x)=\max\left\{\left|x_{i}\right|,1\leq i\leq m\right\}$
F4	$F_{4}(x)=\sum_{i=1}^{m}\left[x_{i}^{2}-10\cos(2\pi x_{i})+10\right]$
F5	$F_{5}(x)=-20\exp(-0.2\sqrt{\frac{1}{n}\sum_{i=1}^{m}x_{i}^{2}})-\exp(\frac{1}{n}\sum_{i=1}^{m}\cos(2\pi x_{i}))+20+e$
F6	$\begin{array}{l} F_{6}(x)=\frac{1}{10}\sum_{i=1}^{m-1}{\left(x_{i}-1\right)}^{2}\left[\sin^{2}+(3\pi x_{i+1})+1\right]+\sum_{i=1}^{m}u(x_{i},5,100,4)+\frac{1}{10}\sin^{2}(\pi x_{1})\\ +\frac{1}{10}{\left(x_{m}-1\right)}^{2}\left[\sin^{2}(2\pi x_{i+1})+1\right] \end{array}$

**Table 2 entropy-24-00048-t002:** Test results.

Function Name	Dimension	Maximum Position	Minimum Position	Optimal Value	Algorithm							
					PSO	X	wPSO	X	wdPSO	X	SPPSO	X
F1	80	100	−100	0	2.60468	107	2.18797	85	$8.40\times 10^{-33}$	37	$4.57\times 10^{-47}$	31
F2	80	10	−10	0	12.8512	163	10.8976	79	$1.39\times 10^{-17}$	86	$5.35\times 10^{-35}$	64
F3	80	100	−100	0	0.53274	109	0.33094	89	$2.68\times 10^{-23}$	87	$4.26\times 10^{-56}$	61
F4	80	5.12	−5.12	0	14.4857	191	9.94959	51	35.26952	40	0	28
F5	80	32	−32	0	3.2367	138	2.50804	80	$8.88\times 10^{-16}$	112	$2.65\times 10^{-23}$	30
F6	80	50	50	0	76.6912	143	68.1844	69	222.093	7	0	12

**Table 3 entropy-24-00048-t003:** Algorithm steps.

Step	Content
Step 1	Set the given value $y_{r}$, initialize the parameters and storage variables, and calculate the prediction model Formula (11).
Step 2	Use IGPC to solve the control increment Δu, and judge whether SPPSO is used for optimization according to the Δu constraint range. If the result is Yes, go to Step 3; if the result is No, go to Step 4.
Step 3	Initialize the population, set Equation (15) as the fitness function for optimization.
Step 4	Work out the system input u(t) according to equation (19), and then calculate the output y(t).
Step 5	Update the status storage sequence and repeat steps two through five until control is complete.

## Data Availability

Not Applicable.

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
