# Peer review of "Improved Particle Swarm Optimization Based on Entropy and Its Application in Implicit Generalized Predictive Control"

_entropy, 2021, doi:10.3390/e24010048_

Round 1

Reviewer 1 Report

The article's authors proposed a modification of the particle swarm optimization (PSO) method and applied it to predictive control.

Advantages of the article:

1) The authors proposed a novel modification of PSO.

2) The authors implemented the modification and compared it with other modified versions of PSO.

3) The authors applied the methodology to the predictive control paradigm.

Disadvantages of the article:

1) The authors show in Table 2 the test results only for dimensionality 80. What about other values?

2) The authors did not compare results of predictive controller optimization with results obtained by other methods known from the literature.

3) Minor typo errors in equations - for example, "*" character as multiplication char. 

Author Response

Thank you for your patience and preciseness. Please see the attachment.

Reviewer 2 Report

This paper is about a new improved particle swarm optimization (PSO) algorithm proposed by the authors to solve constraint problems.

  Several suggestions for improving the manuscript.

#1: In the abstract, the authors use the term PSO-IGPC without explaining it.

#2: Introduction needs to be revised, e.g., the authors use the term "MPPSO" (line 34) without explaining it. The introduction should also present other common approaches similar and/or alternative to PSO. Also, it should be properly explained teh SR term in the "this paper combines SR with PSO" (line 41). The introduction should explain more deeply the implicit Generalized Predictive Control problem.

#3: In line 48 the terms MAC, DMC and CARIMA, PID (line 56) are used and not properly explained. All the acronyms used in the manuscript should be clearly presented.   

#4: The GPC paragraphs should be in the beginning of the Introduction since they represent the area where the algorithm will focus on.  

#5: Explain why the "After analysis, it is found that the velocity term is unnecessary." (line 104) using other references (Are there any?).   

#6: In section 2.3, authors use the term SR (line 116 and 121). It should be explained and references should be provided.  

#7: In line 130 authors write "in Figure 1 w1". The W1 series be mentioned differently? In Figure 1, the axis should be explained (w and t). Suggestion: use Figure 1 (w1), etc.

#8: In line 158, text uses a different font. I would recommend avoiding the term "subsection". Use "section". Apply this recommendation to all document.  

#9: The variables from the formulas should be used in the text formatted in italic.

#10: Best results from Table 2 should be highlighted. Also, the legend of Table 1 and 2 should contain more information. Table 2 font text is different.  

#11: Figure 5 needs to be more explained.   

#12: Section 3.2.1 and 3.2.2 should be combined and the heading removed. Also, the content (steps) of the algorithm should be formatted as an algorithm and explained making the necessary references to the flow chart. 

#13: section 4 should contain more results.  

#14: Conclusions are too short. They need to be properly explained.    

Author Response

(The authors gave the same response as above.)

Round 2

Reviewer 2 Report

The authors have followed the reviewer suggestion. They have significantly improved the overall manuscript and tried to incorporate almost all the provided suggestions. Regarding the overall merit of the manuscript, they have an interesting work and tried to explain it in the publication. The area where they applied the algorithm and its challenges are not easy to understand but they have tried to incorporate the requested changes. 

Author Response

Thank you very much for your patience and preciseness. We have revised the introduction of the article according to your suggestion. We will also strengthen our study of English in our future research work.